# A Single Image 3D Reconstruction Method Based on a Novel Monocular Vision System

**DOI:** 10.3390/s20247045

**Published:** 2020-12-09

**Authors:** Fupei Wu, Shukai Zhu, Weilin Ye

**Affiliations:** 1Department of Mechanical Engineering, Shantou University, Shantou 515063, China; fpwu@stu.edu.cn; 2Key Laboratory of Intelligent Manufacturing Technology, Ministry of Education, Shantou University, Shantou 515063, China; 14skzhu@stu.edu.cn

**Keywords:** machine vision, 3D measurement, monocular vision system, automatic optical inspection

## Abstract

Three-dimensional (3D) reconstruction and measurement are popular techniques in precision manufacturing processes. In this manuscript, a single image 3D reconstruction method is proposed based on a novel monocular vision system, which includes a three-level charge coupled device (3-CCD) camera and a ring structured multi-color light emitting diode (LED) illumination. Firstly, a procedure for the calibration of the illumination’s parameters, including LEDs’ mounted angles, distribution density and incident angles, is proposed. Secondly, the incident light information, the color distribution information and gray level information are extracted from the acquired image, and the 3D reconstruction model is built based on the camera imaging model. Thirdly, the surface height information of the detected object within the field of view is computed based on the built model. The proposed method aims at solving the uncertainty and the slow convergence issues arising in 3D surface topography reconstruction using current shape-from-shading (SFS) methods. Three-dimensional reconstruction experimental tests are carried out on convex, concave, angular surfaces and on a mobile subscriber identification module (SIM) card slot, showing relative errors less than 3.6%, respectively. Advantages of the proposed method include a reduced time for 3D surface reconstruction compared to other methods, demonstrating good suitability of the proposed method in reconstructing surface 3D morphology.

## 1. Introduction

Reconstructing three-dimensional (3D) surface morphology can help to quantify product surface quality information, characterize the product defects in manufacturing process and analyze the defects distribution. Therefore, 3D reconstruction technology has been widely used in industrial manufacturing processes [1,2,3,4]. With the rapid increase of automation technology, it is necessary to develop fast and reliable methods for 3D reconstruction and measurement of surface morphology to meet stricter process quality requirements [5,6,7].

In terms of 3D reconstruction and measurement, previous works mainly focused on contact and non-contact approaches. Traditional contact approaches, such as coordinate measuring instrument, atomic force microscopy (AFM) and other microcopies [8,9,10], are highly precise in measuring, but slow and costly, therefore not suitable to meet the fast measurement requirements for online fabrication. On the other hand, acoustic detection methods [11,12,13] and magnetic nanoparticle-based detection methods [14] belong to non-contact approaches, which have been extensively used in manufacturing processes [13,14]. Such methods are characterized by a high detection speed while the measurement accuracy is easily affected by the response time and active signal source.

Compared with above mentioned contact and non-contact methods, optical measurement methods are widely used as they are characterized by non-contact, non-damage, high-resolution and high -speed [15,16,17,18]. Multi-vision systems are typical optical measurement methods, able to reconstruct the 3D surface by calibrated cameras and space intersection calculation [19,20,21,22,23]. However, image matching is still a challenging task during the 3D reconstruction surface process because it needs a massive matching computation and it is prone to environment lighting issues [24,25,26], resulting in being highly costly and computationally time consuming [27].

Single image methods have also recently been proposed for 3D surface reconstruction, such as the monocular vision method and the structured light method. Currently, the monocular vision method originally derives from shape-from-shading (SFS) methods [28,29], which estimates the 3D shape from shading, defocus and correspondence using just a single-capture passive light-field image. Because of SFS methods mainly being developing from the Lambertian model of illumination reflection [30], they are ill-conditioned and have no unique solution. To solve such problem, SFS usually assumes that the object studied is a smooth object, and then the regularization model of the SFS problem is established according to the relevant conditions, which mainly include the minimization method, propagation method, local method and linearization method [31,32]. However, experimental results of 3D surface reconstruction are prone to uncertainties [33]. Recently, optical 3D measurement methods have been developed based on the structured light [34,35]. In such methods, color evaluation is a crucial aspect. To reveal the relationships between color and 3D reconstruction, the color calibration is usually formulated as a color pureness problem [36,37,38]; nevertheless, the color pureness can easily lead to the need for more accurate calibration.

From the available literature, the 3D measurement methods have developed quickly in recent years, showing an increasing trend in terms of computational speed and reliability [39]. However, the implementation of existing reconstruction methods still cannot meet fast and reliable measurement requirements. This paper aims at developing a quick 3D reconstruction method based on a monocular vision system, in which ring structured light emitting diode (LED) illumination is used to project the object’s 3D information into the image and only one image is required to realize the 3D measurement process.

This paper is organized as follows. Monocular vision system and its image information are presented in Section 2. Then, the imaging principle is analyzed, the model of 3D reconstruction is built, and the proposed method is developed in Section 3. Following that, a simplified calculation method for the proposed algorithm is designed in Section 4, experimental results are presented and discussed in Section 5, while conclusions are reported in Section 6.

## 2. Monocular Vision System and its Image Information

The proposed monocular vision system is composed of a three-level charge coupled device (3-CCD) camera and a red-green-blue (RGB) ring illumination source. As shown in Figure 1, the color light source includes three circular LED rings, respectively red, green and blue. The light source parameters, including sizes, position, incident angles and distribution density of LEDs are designed and calibrated accurately [40]. The detected objects are placed in the center of the worktable. In the acquired color image, each pixel has three intensity values of red, green and blue. Furthermore, these gray values vary with inclination of the detected surface due to light incident angles. Thus, the intensity values of red, green and blue reflect the variation trend of detected surface, in other words, the red, green and blue irradiates to the flat, slow slant and rapid slant surface, respectively [41]. Such characteristics will be used for image regions segmentation and to design the proposed algorithm.

Unlike the SFS methods, the proposed method uses a multi-color ring structured light, which has calibrated parameters in terms of height, incident angle and distribution density for each LED, in order to get the incident light position information of the acquired color image. Following the calibration, the same incident light intensities will have the same gray values in the same detected regions [40].

## 3. Analysis of Imaging Principle and Model of 3D Reconstruction

### 3.1. Principle of the Imaging based on the Monocular Vision
System and the Ring Structure Light Source

In this section, a 3D reconstruction module is presented based on a monocular vision system. A simplified monocular vision system is shown in Figure 2a, which includes a 3-CCD camera, a lens and a three-color light source (red, green and blue). The three lighting points, namely, $S_{r}$,
$S_{g}$ and
$S_{b}$
represent the red, green and blue luminous points from the ring light source, respectively, all located in the XOZ plane. The detected object cross section *AGB* is also in the XOZ plane. Because of the symmetry properties of the monocular vision system, every cross section has the same imaging process, including the Z-axis and perpendicular to the plane of XOY. In this case, the imaging plane of XOZ is shown in Figure 2b, where $L_{O_{M}M_{H}}$
represents the image plane of the 3-CCD camera, $L_{L_{1}L_{2}}$
is the lens width, the curve
$L_{AGB}$
stands for the detected surface and the point *G* is irradiated by light sources $S_{r}$, $S_{g}$
and $S_{b}$. The point $M_{H}\left(G_{r},G_{g},G_{b}\right)$
is the image of the detected point *G*, along with the gray values of red, green and blue ($G_{r},G_{g},G_{b}$),
respectively. The coordinate value of *G* is defined as ($x_{G},h$
), in other words, the height of the detected point *G* is defined as *h*, *h* is also named $z_{G}$.

In particular, *h* is unknown before reconstruction and measurement. For a focused optical imaging with object distance *H − h*, image distance *d* and a thin lens with focal length *f*, the following relationship holds:
(1)$$\frac{1}{f}=\frac{1}{H-h}+\frac{1}{d}$$

From Equation (1), the detected height can be expressed as:(2)$$h=H-\frac{fd}{d-f}$$


As it is affected by depth of field, *h* may not be the exact height of the detected point. In other words, usually, the image plane is invariable in an optical system, while the object plane is variable due to the surface variety of the detected object within a tolerated range. If the object plane does not match the image plane in an optical system, all the objects located at different distances from the camera will appear blurred [42].
As shown in Figure 3 A, B and C are the detected points, L_1_L_2_ is the lens width and O_M_A_1_ is the image sensor. The detected point A is captured sharply as A_1_ is on the image plane O_M_A_1_, while the detected points B and C are imaged to B_1_ and C_1_, respectively. However, in an optical system, since $A_{1}$
is fixed, both B
and C will be captured as blurred points. Therefore, in this configuration, it is not suitable to calculate the detected height only by Equation (2). To get the exact detected height, additional constraints should be considered.

### 3.2. The 3D Reconstruction Model

In an optical imaging system, every pixel coming from image sensor elements represents a special region in the detected object surface, denoted by $\sigma_{G}$. Its corresponding gray value indicates the energy intensity of the incoming light, coming from the light source and reflected by the special region $\sigma_{G}$ of the detected object.

As shown in Figure 2, the physical point *G*, which is imaged as pixel *M_H_*, is lighted by the light source *S_r_*, *S_g_*, and S_b_, and its gray values are $G_{r},G_{g}$
and $G_{b}$, respectively. The angle of the tangent plane of point *G* to the plane XOY is denoted by *θ*. The angle of incidence of $S_{g}$
to the plane XOY is denoted by $\beta_{g}$, its mirror reflecting line is denoted by $L_{GM_{H}}$
and its reflection angle is denoted by *φ*, representing the theoretical reflecting line to the pixel *M_H_*. The actual reflecting line is denoted by $L_{GR}$
and its angle to the line $L_{GM_{H}}$
is denoted by $\gamma_{g}$. Then, the parallel light reflecting from *G* is denoted by $L_{GM_{H}}$
focusing on the point *M_H_*. As shown in Figure 2, the incoming red, green and blue angles are denoted by $\beta_{r},\beta_{g}$
and $\beta_{b}$, respectively, therefore the following equations hold:(3)$$tan\beta_{r}=\left|\frac{z_{r}-z_{G}}{x_{r}-x_{G}}\right|$$
(4)$$\tan\beta_{g}=\left|\frac{z_{g}-z_{G}}{x_{g}-x_{G}}\right|$$
(5)$$\tan\beta_{b}=\left|\frac{z_{b}-z_{G}}{x_{b}-x_{G}}\right|$$
(6)$$tan\left(\emptyset +\theta \right)=\left|\frac{z_{o}-z_{G}}{x_{o}-x_{G}}\right|$$

In which $\left(x_{r},{\;z}_{r}\right)$, $\;\left(x_{g},{\;z}_{g}\right)$, $\left(x_{b},{\;z}_{b}\right)$, and $\left(x_{o},{\;z}_{o}\right)$
can be measured on the vision system, and $x_{G}$
can be retrieved from its nominal physical size based on its pixel coordinate. Therefore, only the three variables $z_{G},\;\emptyset$
and  θ are unknown in Equations (3)–(6).
Based on the triangle transformation, reflection angles between the theoretical reflecting line and the actual reflecting lines of red, green, and blue, denoted by $\gamma_{r},\gamma_{g}\;$
and $\gamma_{b}$, respectively, which can be expressed as:(7)$$\gamma_{r}=\left|\left(\pi +\theta -\emptyset \right)-\left(\pi -arctan\left|\frac{z_{r}-z_{G}}{x_{r}-x_{G}}\right|\right)\right|=\left|\theta -\emptyset +arctan\left|\frac{z_{r}-z_{G}}{x_{r}-x_{G}}\right|\right|$$
(8)$$\gamma_{g}=\left|\theta -\emptyset +arctan\left|\frac{z_{g}-z_{G}}{x_{g}-x_{G}}\right|\right|$$
(9)$$\gamma_{b}=\left|\theta -\emptyset +\arctan\left|\frac{z_{b}-z_{G}}{x_{b}-x_{G}}\right|\right|.\;$$

In which $0<\gamma_{r},\gamma_{g},\gamma_{b}<\frac{\pi }{2}$.

The incoming red, green and blue light intensities are indicated as $I_{r},\;I_{g},\;\mathrm{and}\;I_{b}$, respectively. Such intensities are reflected by the detected surface $\sigma_{G}$, and part of their energy is transmitted to the pixel element $M_{H}$. Furthermore, $I_{r},\;I_{g}\;\mathrm{and}\;I_{b}$
are affected by the detected material and the inclination angle of the detected surface. Based on Equations (7)–(9) and the reference model proposed in [40], if the influencing factors on the material to red, green and blue light are denoted by $f_{r},\;f_{g}\;\mathrm{and}\;f_{b}$,
respectively, which can be obtained by experimental tests, then the relationships between incoming lights and their gray values can be built as follows:(10)$$G_{r}=I_{r}\cdot \sigma_{G}\cdot f_{r}\cdot \cos\theta \cdot {\cos\gamma }_{r}$$
(11)$${\;G}_{g}=I_{g}\cdot \sigma_{G}\cdot f_{g}\cdot \cos\theta \cdot {\cos\gamma }_{g}$$
(12)$$G_{b}=I_{b}\cdot \sigma_{G}\cdot f_{b}\cdot \cos\theta \cdot {\cos\gamma }_{b}$$

Inserting Equations (7)–(9) into Equations (10)–(12) results in
(13)$$G_{r}=I_{r}\cdot \sigma_{G}\cdot f_{r}\cdot cos\theta \cdot cos\left|\theta -\emptyset +arctan\left|\frac{z_{r}-z_{G}}{x_{r}-x_{G}}\right|\right|=I_{r}\cdot \sigma_{G}\cdot f_{r}\cdot cos\theta \cdot cos\left(\theta -\emptyset +arctan\left|\frac{z_{r}-z_{G}}{x_{r}-x_{G}}\right|\right)$$

Similarly,
(14)$$G_{g}=I_{g}\cdot \sigma_{G}\cdot f_{g}\cdot cos\theta \cdot cos\left|\theta -\emptyset +arctan\left|\frac{z_{g}-z_{G}}{x_{g}-x_{G}}\right|\right|$$
(15)$$G_{b}=I_{b}\cdot \sigma_{G}\cdot f_{b}\cdot \cos\theta \cdot \cos\left|\theta -\emptyset +\arctan\left|\frac{z_{b}-z_{G}}{x_{b}-x_{G}}\right|\right|$$

In which, $G_{r},\;G_{g},\;\mathrm{and}\;G_{b}$
are the gray values of red, green and blue obtainable from the detected color image. $I_{r},\;I_{g},I_{b}$, $\sigma_{G},\;$, $f_{r},\;f_{g}$
and $f_{b}$, can be measured by calibrated experimental tests. In other words, there are only three variables of $z_{G},\emptyset ,\;\theta$
in Equations (13)–(15). Therefore, $z_{G}$
can be calculated by solving Equations (13)–(15), which indicate the height information of the detected area $\sigma_{G}$. By repeating this procedure across the whole detected area, the whole detected surface will be reconstructed, and the measurement of surface size can be carried out.

## 4. Simplified Calculation Method for the Proposed Algorithm

As mentioned in Section 3, in order to reconstruct the whole detected surface, the proposed method needs to calculate all the pixels of the detected image by Equations (13)–(15), being very inefficient and time consuming. In this section, the proposed algorithm is optimized to improve its reconstruction speed while its inspection accuracy remains controllable.

Two aspects will be considered in the following to simplify the proposed algorithm. Firstly, the detected color image is divided into different regions based on the color gradients. Secondly, polar coordinates are used to simplify the computing processing based on the cycle calculation.

### 4.1. Algorithm Simplified Based on Regions

Within the uniform material, the surface of the detected object has the same property of the specular and diffuse reflection, which abides to the law of light reflection. In this manuscript, the detected image is obtained by the monocular vision system, which includes a 3-CCD color digital camera and a 3-color (red, green and blue) ring structured light source. As shown in Figure 1 and analyzed in Section 1, the red, green and blue distribution in the detected image indicates the surface slant gradients of the detected object [42]. Within the single-color layer image, the same gray values, which are adjacent, indicate the same height in the detected surface, and the increased or decreased gray values indicate variations in the detected surface height. Therefore, the detected image can be divided into several regions based on the gray levels, and the height information of every region can be calculated by Equations (13)–(15) using only one-pixel information within the same regions. In this way, a high number of redundant calculation tasks will be avoided, speeding up the reconstruction process.

### 4.2. Algorithm Simplified based on Coordinates Transformation

As shown in Figure 4, supposing that the detected point *G* corresponds to the pixel coordinates (*i*,*j*) in the detected image. $\delta_{x},{\;\delta }_{y}$
is denoted as the ratio factor between the physical region and its pixel’s region in the direction of *X* and *Y*, respectively. In the plane of XOY, the region of $\left(i\cdot \delta_{x},\;i\cdot \delta_{x}+\delta_{x}\right)$
in width and $\left(i\cdot \delta_{y},\;i\cdot \delta_{y}+\delta_{y}\right)$
in height, which corresponds to the pixel coordinates (*i*,*j*), is the projection of the detected surface region *G*.
$z_{G}$
is the height of the detected surface region *G*. Then
(16)$$r=\sqrt{{\left(i\cdot \delta_{x}+0.5\delta_{x}\right)}^{2}+{\left(j\cdot \delta_{y}+0.5\delta_{y}\right)}^{2}}$$
(17)x=r·cosω
(18)y=r·sinω
(19)$$\omega =arcsin\frac{j\cdot \delta_{y}}{r}$$

Inserting Equations (16)–(18) into Equations (13)–(15) and solving the resulting equation for *z* yields:(20)$$z=f\left(x,y\right)=f\left(r\cdot cor\omega ,\;r\cdot sin\omega \right)$$

Then, the detected surface can be expressed as
(21)$$a=1,z=\int_{r=0}^{r_{max}}\int_{\omega =0}^{2\pi }f\left(r\cdot cor\omega ,\;r\cdot sin\omega \right)=\overset{r_{max}}{\underset{r=0}{\sum }}\overset{2\pi }{\underset{\omega =0}{\sum }}f\left(r\cdot cor\omega ,\;r\cdot sin\omega \right)$$

In which, $r_{max}=\sqrt{{\left(i_{max}\cdot \delta_{x}+0.5\delta_{x}\right)}^{2}+{\left(j_{max}\cdot \delta_{y}+0.5\delta_{y}\right)}^{2}}$, and $i_{max}\;,j_{max}$
are limited by the field of view. *z* is calculated based on its pixel in the field of view one by one, then Equation (20) can be expressed as:(22)$$z=\overset{r_{max}}{\underset{r=0}{\sum }}\overset{2\pi }{\underset{\omega =0}{\sum }}f\left(r\cdot cor\omega ,\;r\cdot sin\omega \right)$$

From Equation (21), it indicates that the detected surface can be reconstructed completely by a series of cyclical calculation, which obviously reduces the complexity of computation and improves the efficiency of the 3D reconstruction process.

Therefore, the flow chart of the proposed algorithm can be presented as Figure 5. Especially, the boundary condition is limited to the field of view, and the calculation sequence comes from divided regions and the order of regions.

## 5. Experiments Results and Analysis

The process of three-dimensional measurement is carried out firstly by reconstructing the 3D surface of the detected object, and then by measuring the 3D size information of any point by the optical system. In this section, the 3D measurement process is illustrated according to the proposed method, including its performance evaluation via benchmarking. As shown in Figure 6, the image acquisition system is composed of a JAI CV-M9CL (JAI Corp., Copenhagen, CO, Denmark) 3-CCD camera and the designed light source, (reported in Figure 1). In this case study, the object distance *H* is 285.4 mm and the image distance *d* is 31.6 mm, the focal length *f* is 28.4 mm and the field of view is 24 mm × 18 mm. During the experiments, a noise environment less than 90 dB and stable lighting environment are necessary, the vibration frequency of the worktable should be less than 100 Hz and its amplitude should be less than 0.5 µm, moreover, the light intensity deviation of the light source should be less than 5% within the field of view. In addition, a 3D measurement system VR 5200 (KEYENCE Corp., Osaka, Japan) is used in the experiment (Figure 7) to measure the true height of samples. VR 5200 has 0.1 µm display resolution and 120 × resolution on 15 inch displays is 2.5 mm × 1.9 mm, and its precision of measurement repeatability is less than 0.4 µm without z-connection. Therefore, the object height can be measured accurately by VR 5200 to evaluate the performance of the proposed method comparing with other similar methods.

The process of measuring surface height with the proposed method is shown in Figure 8. Firstly, the working environment should be carefully checked, and the light source must be calibrated, then the sample color images can be acquired by the proposed system, for example, the color image shown in Figure 8a. Secondly, the acquired color image is converted to gray levels for red, green and blue channels, and then gray images are segmented into several regions based on similar gray levels, as shown in Figure 8b,c. Thirdly, each region is located to define the calculation region, and arrangement relations are identified to form the calculation sequence. Additionally, surface height information is cyclically calculated using Equation (22). Lastly, calculation results are arranged based on location information and then the 3D surface can be retrieved as shown in Figure 8d.

To evaluate the performance of the proposed method, four different samples, namely, convex surface, concave surface, angular surface and convex and concave surface were produced by 3D printing using a single material, and tested based on the proposed algorithm, as shown in Table 1. In addition, a subscriber identification module (SIM) mobile card slot was also tested. Furthermore, benchmarking was carried out to compare the proposed method to alternative methods, in terms of reconstruction accuracy and speed.

As shown in Table 1, the four samples in the first row are obtained by utilizing a digital camera and natural light. The second row shows images acquired using the proposed vision system. The bottom row shows the reconstruction results obtained by the proposed algorithm. In order to compare the proposed method to other methods (proposed in [9,29,39]), four section views were considered from the reconstruction results, as shown in Figure 9, Figure 10, Figure 11 and Figure 12, respectively. In these charts, the black line refers to the object height measured by VR 5200, the blue line refers to the object height measured by the method of light microscopy axial-view imaging [9], the yellow line refers to the reconstruction height based on the proposed method, the purple line refers to the reconstruction height by [29] and the green line refers to the reconstruction height by [39].

As regards the convex surface, Figure 9 shows that the proposed reconstruction height curve fluctuates along the actual object height curve with an error less than 2.3%, while the other two methods, respectively proposed by [29] and [39], both have an error over 6%. This result demonstrates that the proposed algorithm can reconstruct and measure the convex surface effectively. Within the error range, the proposed reconstruction height curve is also characterized by a smaller deviation compared to the other methods. This can be explained by the processed gray region segmentation. In addition, the wave-like pattern in the reconstruction height curve, suggests that the uniformity of the light source raying is very important in the proposed algorithm.

As regards the concave surface, Figure 10 shows that the proposed reconstruction height curve fluctuates along the actual object height curve with an error less than 3.2%, showing higher accuracy compared to the other two methods [29] and [39]. This result indicates that the proposed algorithm can also reconstruct and measure the concave surface effectively. Figure 10 also shows that the reconstruction height curve overestimates the actual object height curve. Such error is mainly caused by intensity calibration issues with the incoming light. Such errors can be reduced by properly calibrating the light source parameters.

As regards the angular surface, Figure 11 shows that the proposed reconstruction height curve fluctuates along the actual object height curve with an error less than 3.2%, showing a good suitability of the proposed algorithm in effectively reconstructing the angular surface. The error, shown in Figure 11, indicates that the gray region segmentation and the calibration accuracy of light source parameters are crucial aspects for improving the reconstruction precision with the proposed algorithm.

As regards the convex and concave surface, Figure 12 shows that the proposed reconstruction height curve fluctuates along the actual object height curve with an error less than 2.7%. Such results suggest that the proposed algorithm can be used to reconstruct and measure different types of surfaces, which represent a common scenario for industrial applications.

Furthermore, to evaluate the performance of the proposed algorithm, a mobile SIM card slot was also measured experimentally. The SIM card slot image is shown in Figure 13, and its concave-convex components are highlighted in Figure 13b. The reconstruction results using the proposed method are shown in Figure 14. A comparison with the light microscopy axial-view imaging method [9] and the true height of the SIM card measured by VR-5200 are presented in Figure 15. The chart shows how the proposed method yields very similar results to the method developed by Guo et al. [9] in terms of measurement accuracy. It is also found that the maximum deviation (within 3.6%) corresponds with the bottom of the concave region, this is due to the lower image definition in the region. Adjusting focal length and acquiring clear images can improve measurement accuracy.

To verify the computational load of the proposed method, the SIM card slot is detected with different reconstruction methods. As shown in Figure 16, with an increasing number of SIM card slots, the proposed method (before the simplification procedure) has the slowest detection speed because it needs to compute every pixel to retrieve the height information. The other three methods proposed in [9,29,39] have an intermediate velocity to compute the height of SIM card slots. In contrast, the proposed method after simplification has the fastest reconstruct speed, because it only needs to calculate few pixels within the same gray values of adjacent pixels, and because of the polar coordinate symmetry of monocular vision system, it reconstructs the surface height by a series of cyclical calculations with a double loop program. Such results indicate that the proposed method after simplification can reduce the computation complexity and improve the efficiency of the 3D reconstruction process.

## 6. Conclusions

In this paper, a single image 3D reconstruction method is proposed based on a novel monocular vision system. In such a proposed method, the relationship model of the detected height information and its image gray value is built, and the 3D reconstruction method is presented.

A simplified calculation method is described to speed up the process of 3D reconstruction based on gray regions segmentation and the coordinate transformation.

Experimental results show that the proposed algorithm can reconstruct the 3D size of convex, concave and angular surfaces with errors less than 3.2%. A mobile SIM card slot was also investigated and the resulting measurement error was less than 3.6%, which illustrates the validity of the proposed algorithm. In terms of applicability, the proposed method can be effectively utilized to reconstruct diffuse surfaces, while a low accuracy has been obtained for specular surfaces. In addition, a light source calibration procedure should be carried out prior to detection operations in order to improve the results accuracy.

## Figures and Tables

**Figure 1 sensors-20-07045-f001:**
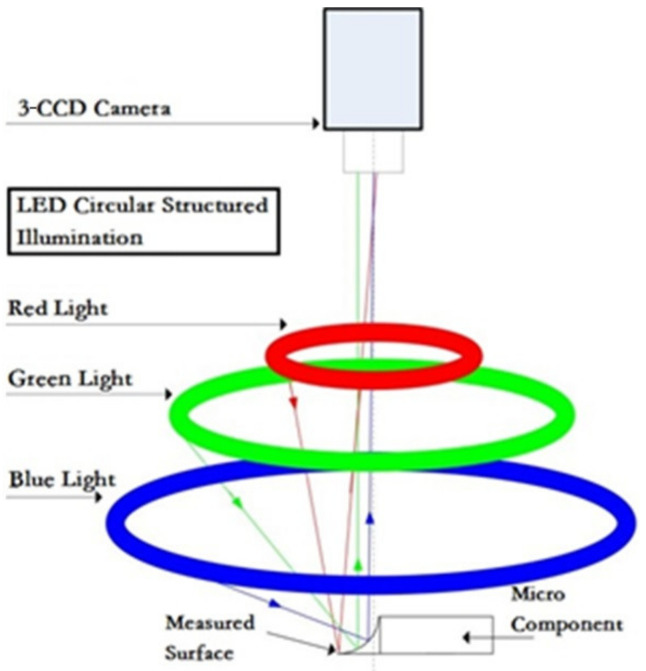
The monocular vision system.

**Figure 2 sensors-20-07045-f002:**
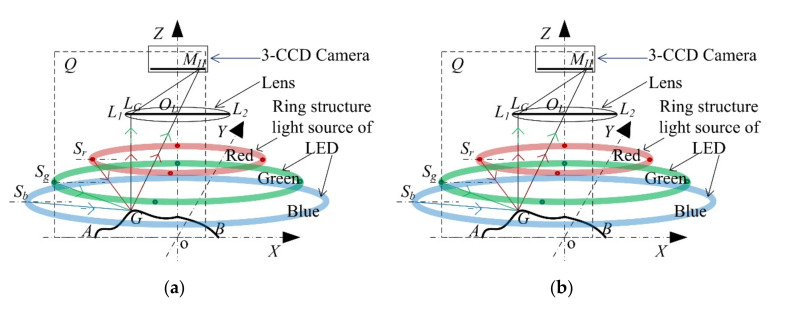
(**a**) The monocular vision system; (**b**) the imaging plane of XOZ.

**Figure 3 sensors-20-07045-f003:**
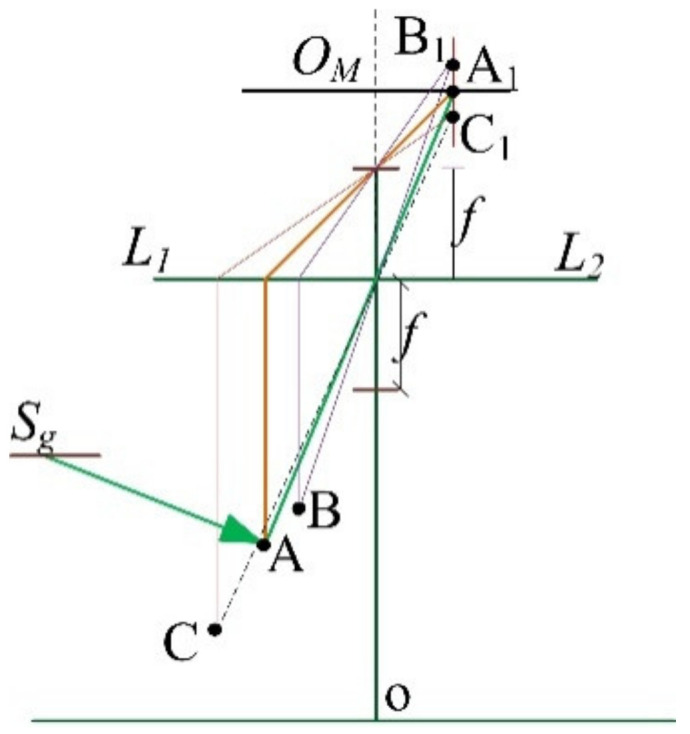
The imaging process of the measured object.

**Figure 4 sensors-20-07045-f004:**
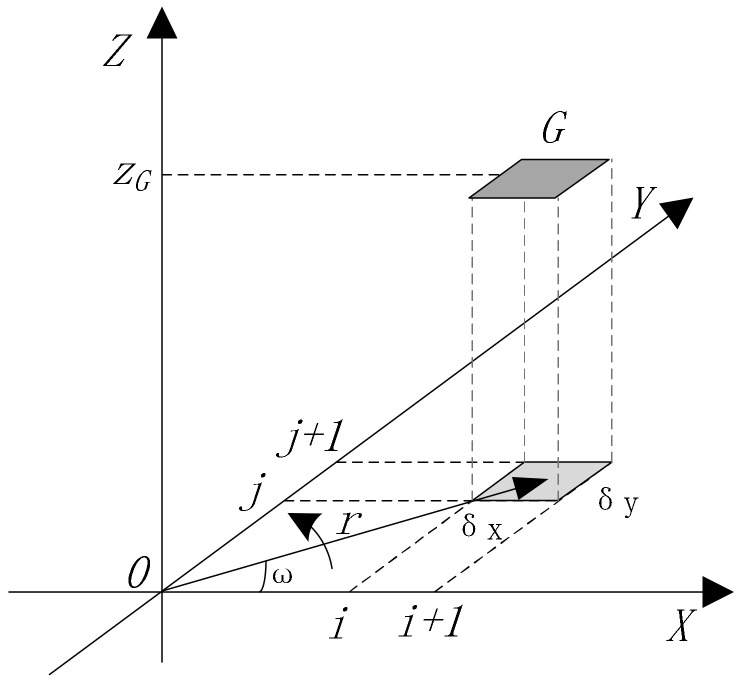
Algorithm simplification based on coordinates transformation.

**Figure 5 sensors-20-07045-f005:**
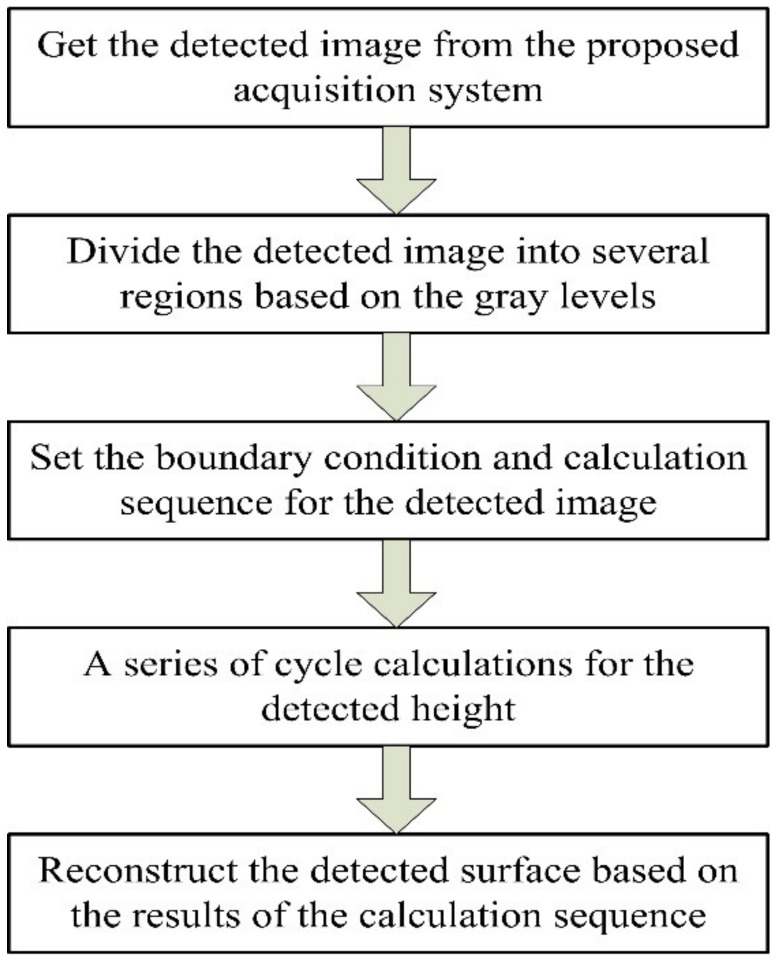
The flow chart of the proposed algorithm.

**Figure 6 sensors-20-07045-f006:**
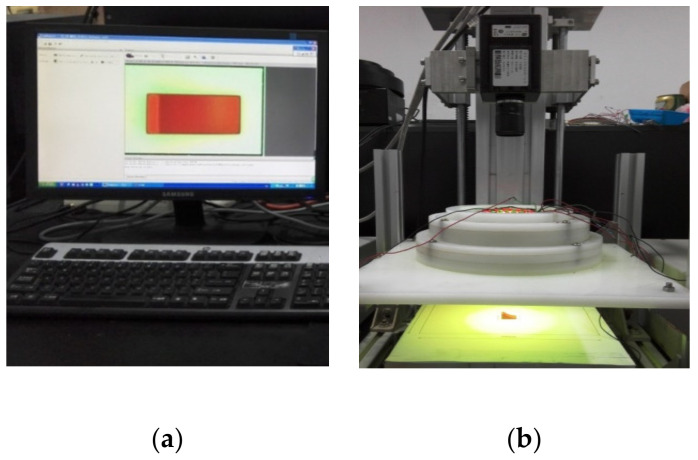
(**a**) The acquisition image; (**b**) The experimental platform.

**Figure 7 sensors-20-07045-f007:**
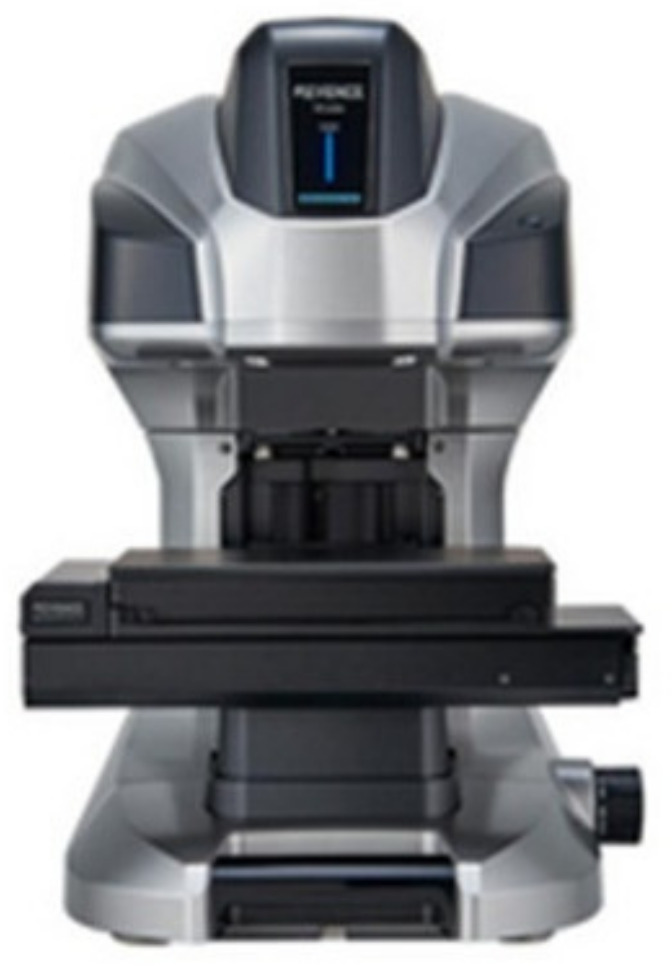
3D measurement system VR 5200.

**Figure 8 sensors-20-07045-f008:**
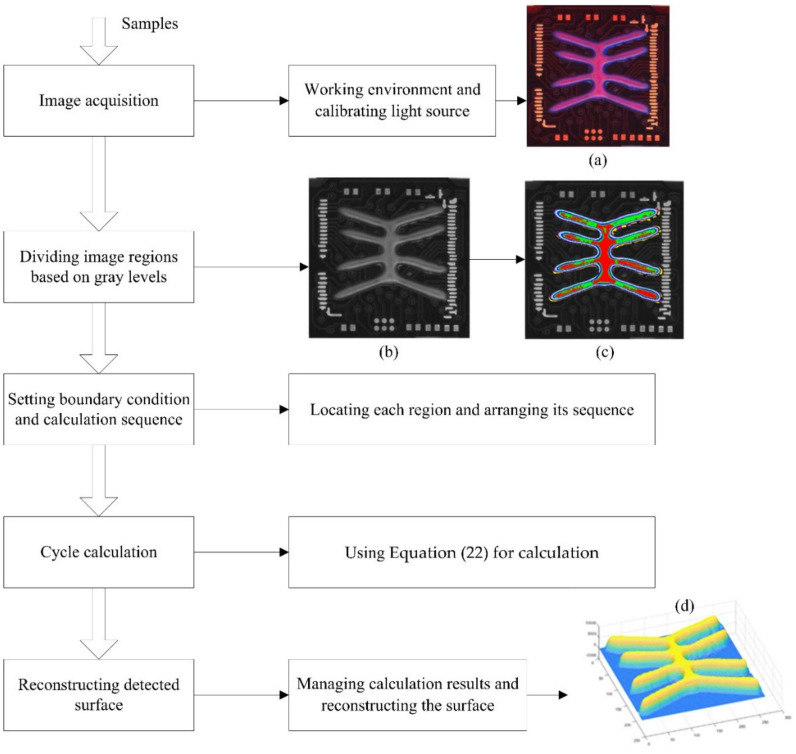
Surface height measurement process scheme using the proposed method. (**a**) Color image acquisition; (**b**) grayscale image; (**c**) image regions segmentation and (**d**) detected surface reconstruction.

**Figure 9 sensors-20-07045-f009:**
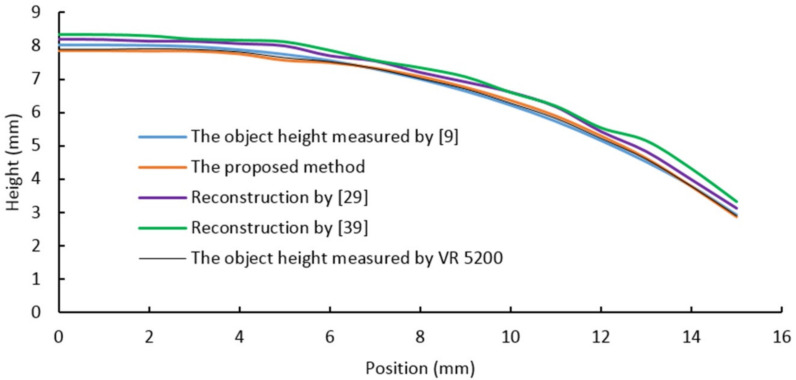
Convex surface reconstruction results.

**Figure 10 sensors-20-07045-f010:**
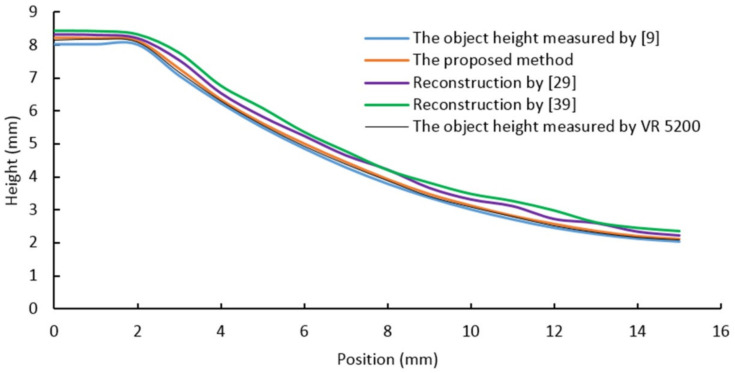
Concave surface reconstruction results.

**Figure 11 sensors-20-07045-f011:**
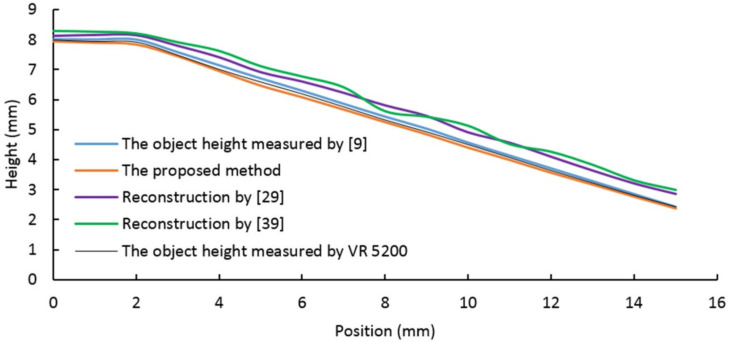
Angular surface reconstruction results.

**Figure 12 sensors-20-07045-f012:**
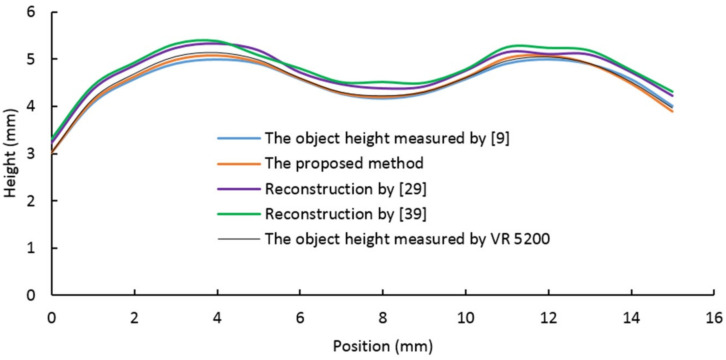
Convex and concave surface reconstruction results.

**Figure 13 sensors-20-07045-f013:**
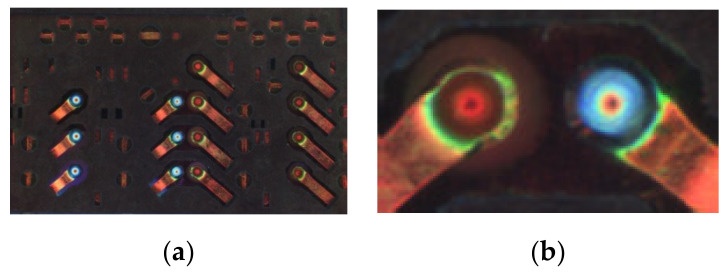
(**a**) SIM card slot acquired image; (**b**) an example of concave-convex components.

**Figure 14 sensors-20-07045-f014:**
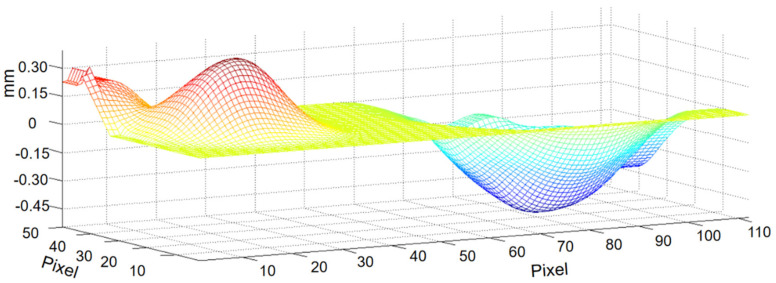
Mobile SIM card slot reconstruction result via the proposed method.

**Figure 15 sensors-20-07045-f015:**
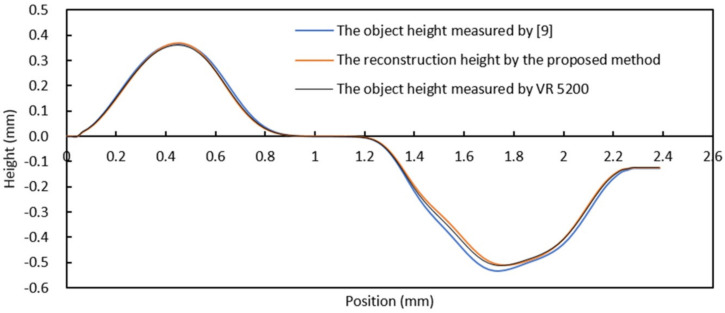
Measurement results comparison.

**Figure 16 sensors-20-07045-f016:**
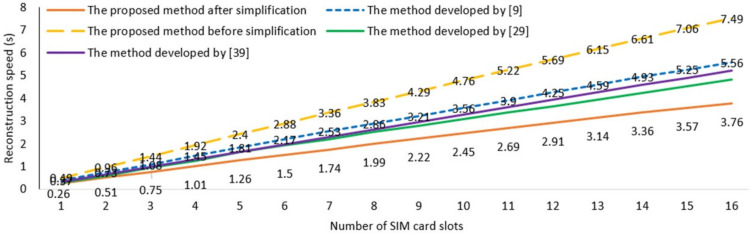
The inspection speed with different methods.

**Table 1 sensors-20-07045-t001:** Experimental results.

	Convex Surface Sample	Concave Surface Sample	Angular Surface Sample	Convex and Concave Surface Sample
Sample images	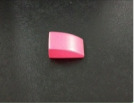	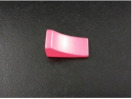	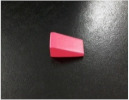	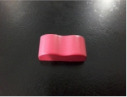
Images acquired via the proposed system	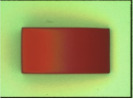	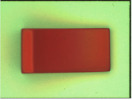	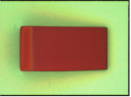	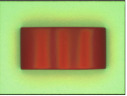
Reconstructed images via the proposed method	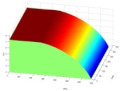	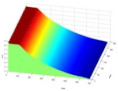	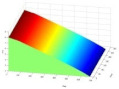	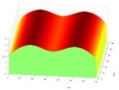

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
