# Peer review of "A Single Image 3D Reconstruction Method Based on a Novel Monocular Vision System"

_sensors, 2020, doi:10.3390/s20247045_

Round 1
Reviewer 1 Report
- This paper presents a single image 3D reconstruction method based on 3-CCD camera and a multi-color LED ring arrays structure illumination.
- In the introduction, there is too much information on the previous work about this research. Summarize them more concisely or write them in the related work section.
- In the experiment section, there should be an explanation about how to measure the object height accurately to evaluate the performance of the proposed method. I think this point is very crucial in the next round of review.
- In the experiment section, only the inspection speed of the proposed method is compared with other methods. The authors should compare 3D reconstruction results to other methods. This point will also require major revisions.
- The author should write full words before using the abbreviation.
- The manuscript needs to be professionally edited by a native speaker.
Author Response
This paper presents a single image 3D reconstruction method based on 3-CCD camera and a multi-color LED ring arrays structure illumination.
- In the introduction, there is too much information on the previous work about this research. Summarize them more concisely or write them in the related work section.
- The introduction has been reduced to better summarize previous works as per reviewer’s comment. Please refer to lines 45 - 65 in the revised manuscript.
- In the experiment section, there should be an explanation about how to measure the object height accurately to evaluate the performance of the proposed method. I think this point is very crucial in the next round of review.
- An explanation about how to measure the object height accurately has been added in section 5 (lines 246-254 in the revised manuscript), and Figure 7 has been added to illustrate such process within the proposed method.
- In the experiment section, only the inspection speed of the proposed method is compared with other methods. The authors should compare 3D reconstruction results to other methods. This point will also require major revisions.
- In the experiment, two other similar methods (referenced in [29] and [39]) have been considered for comparison purposes with the proposed method in the revised manuscript. Experimental results are shown in Figure 8 to Figure 11 and Figure 15, to demonstrate the validity of the proposed method.
- The author should write full words before using the abbreviation.
- All the acronyms have been expanded when first mentioned throughout the whole revised manuscript.
- The manuscript needs to be professionally edited by a native speaker
- The whole revised manuscript has been thoroughly edited in order to improve English syntax and grammar.
Reviewer 2 Report
In this manuscript, a single image 3D reconstruction method is proposed based on a novel monocular
vision system, which includes a 3-CCD camera and a multi-color LED ring arrays structure illumination. The design of the device to fulfill the goal of 3D reconstruction is interesting. But the proposed approach has several concerns to be addressed before publishing.
- The environment of the system to operate the device is obviously strictly limited. The authors should have some discussion about how to use this device.
- Similar approaches, i.e. to do 3D reconstruction based on a single RGB or RGB-D image have been extensively studied. The authors should include performance comparison with the state-of-the-arts in order to clarify the significant contributions that the proposed approach brought to the reader.
- Again, the authors should tell us the constraints of the proposed system. I means when the system cannot get accurate results.
Author Response
In this manuscript, a single image 3D reconstruction method is proposed based on a novel monocular vision system, which includes a 3-CCD camera and a multi-color LED ring arrays structure illumination. The design of the device to fulfill the goal of 3D reconstruction is interesting. But the proposed approach has several concerns to be addressed before publishing.
- The environment of the system to operate the device is obviously strictly limited. The authors should have some discussion about how to use this device.
- The environment of the system to operate the device has been described and discussed in section 5 (lines 236 – 239 in the revised manuscript). The surface height measurement process has also been added in section 5 (lines 246 – 254) and shown in Figure 7.
- Similar approaches, i.e. to do 3D reconstruction based on a single RGB or RGB-D image have been extensively studied. The authors should include performance comparison with the state-of-the-arts in order to clarify the significant contributions that the proposed approach brought to the reader.
- In the experiment, two other similar methods (referenced in [29] and [39]) have been considered for comparison purposes with the proposed method in the revised manuscript. Experimental results are shown in Figure 8 to Figure 11 and Figure 15, to demonstrate the validity of the proposed method.
- Again, the authors should tell us the constraints of the proposed system. I means when the system cannot get accurate results.
- The applicability limitations have been briefly described in the conclusion section and reported below for reviewer’s reference
“In terms of applicability, the proposed method can be effectively utilized to reconstruct diffuse surfaces, while a low accuracy has been obtained for specular surfaces. In addition, a light source calibration procedure should be carried out prior to detection operations in order to improve the results accuracy”.
Round 2
Reviewer 1 Report
Overall: Most questions are answered adequately. A remaining comment is given below.
Previous review comment 2:
In the experiment section, there should be an explanation about how to measure the object height accurately to evaluate the performance of the proposed method. I think this point is very crucial in the next round of review.
Your answer:
An explanation about how to measure the object height accurately has been added in section 5 (lines 246-254 in the revised manuscript), and Figure 7 has been added to illustrate such process within the proposed method.
Review comments:
The previous review comment 2 asked to explain how to measure the ground truth (the object height in Figure 14). This has not been addressed adequately. The method for measuring the object height needs to be explained in the paper. This is crucial for the proposed method. Without a satisfactory explanation, this paper needs to be rejected.
Previous review comment 3:
In the experiment section, only the inspection speed of the proposed method is compared with other methods. The authors should compare 3D reconstruction results to other methods. This point will also require major revisions.
Your answer:
In the experiment, two other similar methods (referenced in [29] and [39]) have been considered for comparison purposes with the proposed method in the revised manuscript. Experimental results are shown in Figure 8 to Figure 11 and Figure 15, to demonstrate the validity of the proposed method.
Review comments:
In figure 8 – 11, the object height (the ground truth) should be presented in the same plot to evaluate the performance of the proposed method and to compare it with others. Again, obtaining the right ground truth is the most important requirement in the proposed method. Without this, the paper has no value.
Author Response
Overall: Most questions are answered adequately. A remaining comment is given below.
Previous review comment 2:
In the experiment section, there should be an explanation about how to measure the object height accurately to evaluate the performance of the proposed method. I think this point is very crucial in the next round of review.
Your answer:
An explanation about how to measure the object height accurately has been added in section 5 (lines 246-254 in the revised manuscript), and Figure 7 has been added to illustrate such process within the proposed method.
Review comments:
The previous review comment 2 asked to explain how to measure the ground truth (the object height in Figure 14). This has not been addressed adequately. The method for measuring the object height needs to be explained in the paper. This is crucial for the proposed method. Without a satisfactory explanation, this paper needs to be rejected.
Reply: A 3D measurement system VR 5200 (KEYENCE Corp., Osaka, Japan) is used in the experiment (Figure 7) to measure the true height of samples. VR 5200 has 0.1 um display resolution and 120 × resolution on 15 inch displays is 2.5 mm×1.9 mm, and its precision of measurement repeatability is less than 0.4 um without z-connection. This explanation can been seen in lines 240-245.
Thank you very much for your advice.
Previous review comment 3:
In the experiment section, only the inspection speed of the proposed method is compared with other methods. The authors should compare 3D reconstruction results to other methods. This point will also require major revisions.
Your answer:
In the experiment, two other similar methods (referenced in [29] and [39]) have been considered for comparison purposes with the proposed method in the revised manuscript. Experimental results are shown in Figure 8 to Figure 11 and Figure 15, to demonstrate the validity of the proposed method.
Review comments:
In figure 8 – 11, the object height (the ground truth) should be presented in the same plot to evaluate the performance of the proposed method and to compare it with others. Again, obtaining the right ground truth is the most important requirement in the proposed method. Without this, the paper has no value.
Reply: The true height (based on worktable) of samples and SIM card measured by VR-5200 (KEYENCE Corp., Osaka, Japan). the object height (the ground truth) has been presented in the same plots (Figure 9-12 and 15) to evaluate the performance of the proposed method and to compare it with others.
Thank you very much for your advice.
The new revised manuscript can been seen in this attachment.

Reviewer 2 Report
This paper has been much improved. Thus, it is acceptable in the current form.
Author Response
Reply: Thank you very much for your advice and attention.
The new revised manuscript can been seen in this attachment. Which includes how to measure the object height (the ground truth) accurately to evaluate the performance of the proposed method.
